# MicroRNAs in BM-MSC-Derived Extracellular Vesicles Promote Angiogenesis: An in Vitro Model Study

**DOI:** 10.3390/biomedicines13102353

**Published:** 2025-09-25

**Authors:** Tomomi Kusakabe, Yoshiki Wada, Tomohiro Umezu, Masahiko Kuroda, Hitoshi Okochi, Toshiya Nishibe, Ayako Inoue, Takahiro Ochiya, Shoji Fukuda

**Affiliations:** 1Center for Cell Therapy and Regenerative Medicine, Tokyo Medical University, Tokyo 160-0023, Japan; k_tomomi@tokyo-med.ac.jp (T.K.); wadasrg2@tmd.ac.jp (Y.W.); 2Department of Vascular Surgery, Institute of Science Tokyo, Tokyo 113-8519, Japan; 3Department of Molecular Pathology, Tokyo Medical University, Tokyo 160-8402, Japan; t_umezu@tokyo-med.ac.jp (T.U.); kuroda@tokyo-med.ac.jp (M.K.); 4Department of Regenerative Medicine, Research Institute, National Center for Global Health and Medicine, Tokyo 162-8655, Japan; okochihi@khaki.plala.or.jp; 5Department of Cardiovascular Surgery, Tokyo Medical University, Tokyo 160-0023, Japan; toshiyanishibe@yahoo.co.jp; 6Faculty of Medical Informatics, Hokkaido Information University, Ebetsu 069-8585, Japan; 7Department of Molecular and Cellular Medicine, Institute of Medical Science, Tokyo Medical University, Tokyo 160-0023, Japan; mgrandis1885@yahoo.co.jp (A.I.); tochiya@tokyo-med.ac.jp (T.O.)

**Keywords:** angiogenesis, extracellular vesicles, mesenchymal stromal cell, microRNA

## Abstract

**Background/Objectives:** Critical limb ischemia (CLI) is a severe manifestation of peripheral arterial disease with limited treatment options. Mesenchymal stromal cell (MSC) therapy has shown promise, but variability in efficacy suggests that paracrine mechanisms, particularly extracellular vesicle (EV)-associated microRNAs (miRNAs), may play a central role. **Methods:** We analyzed angiogenesis-related miRNAs in bone marrow-derived MSCs (BM-MSCs) and their EVs. Five angiomiRs (miR-9, miR-105, miR-126, miR-135b, miR-210) were examined; only miR-126, miR-135b, and miR-210 were consistently detected in EVs. Expression variability was assessed across donor age and individuals. Functional evaluation was performed using co-culture of BM-MSCs with human umbilical vein endothelial cells (HUVECs) and by transfecting synthetic miRNAs into HUVECs. Tube formation assays quantified angiogenesis, and angiogenesis-related protein expression (VEGF, FGF, Endoglin, uPA) was analyzed. Biological replicates (multiple donors) and technical replicates (duplicate assays) were clearly defined to ensure reproducibility. **Results:** Co-culture of BM-MSCs and HUVECs significantly enhanced angiogenesis in a dose-dependent manner. EVs selectively packaged angiogenic miRNAs, with expression levels varying according to donor age and inter-individual variability. Transfection of miR-126, miR-135b, and miR-210 individually enhanced tube formation, while the miR-126 + miR-135b combination and triple transfection elicited the strongest effects. Protein analysis confirmed upregulation of VEGF, FGF, and Endoglin. Notably, miR-210 did not further enhance angiogenesis beyond miR-126 + miR-135b but may exert context-dependent effects. **Conclusions:** This study demonstrates that BM-MSC-derived EV miRNAs promote angiogenesis via combinatorial mechanisms, providing mechanistic support for ongoing CLI therapy. Our findings highlight the translational potential of EV-based nucleic acid therapeutics for ischemic disease.

## 1. Introduction

Critical limb ischemia (CLI), a severe manifestation of peripheral arterial disease, continues to present a major clinical challenge due to the scarcity of effective treatment options for patients who are not candidates for surgical or endovascular revascularization. CLI is the most severe form of atherosclerosis obliterans (ASO), characterized by stenosis or occlusion of lower limb arteries due to atherosclerotic plaques, resulting in impaired blood supply [1]. It is associated with increased morbidity and mortality, particularly in patients with comorbidities such as diabetes, ischemic heart disease, and kidney disease [2]. Even with optimal treatment, amputation and mortality rates remain as high as 30% and 25% per year, respectively [3]. Despite advances in revascularization techniques, there is still a strong unmet need for novel regenerative approaches.

In recent years, extracellular vesicles (EVs), including exosomes and microvesicles, have emerged as critical mediators of intercellular communication, particularly in regenerative medicine. EVs derived from mesenchymal stromal cells (MSCs) carry functional cargos such as proteins, mRNAs, and microRNAs (miRNAs), enabling them to influence recipient cells via paracrine signaling pathways [4,5,6]. These findings have shifted the focus from direct engraftment of MSCs toward their secretome and EVs as primary effectors of tissue repair. Indeed, our previous clinical and preclinical studies demonstrated variability in the therapeutic efficacy of MSC transplantation [7,8], which could not be fully explained by cell engraftment, leading us to hypothesize that EV cargo, particularly angiogenesis-related miRNAs, may play a central role. Beyond disease-specific reports, consensus guidelines by the International Society for Extracellular Vesicles (ISEV) provide standardized definitions, reporting criteria, and methodological recommendations for EV research (MISEV2023), which frame the rigor and interpretation of EV studies in regenerative medicine [9]. Foundational reviews also delineate EV biogenesis, cargo selection, and functional delivery to recipient cells, highlighting their relevance as therapeutics and delivery vehicles [10,11].

Among many miRNAs implicated in angiogenesis, miR-126, miR-135b, and miR-210 have been consistently reported to regulate endothelial function and vascular growth:miR-126 maintains vascular integrity by modulating PI3K/AKT and MAPK pathways through repression of Spred1 and PIK3R2 [12,13].miR-135b enhances angiogenesis under hypoxic conditions via HIF-related regulators and VEGF signaling [14,15].miR-210, a “hypoxamiR,” promotes endothelial survival and migration under hypoxia through ephrin-A3 inhibition and VEGF pathway modulation [16].

At a broader level, cardiovascular and endothelial microRNA reviews summarize how angiomiRs coordinate PI3K/AKT, MAPK/ERK, and hypoxia-responsive (HIF-dependent) programs to control endothelial proliferation, migration, and tube formation. These syntheses position miR-126 (Spred1/PIK3R2), miR-135b (HIF axis), and miR-210 (ephrin-A3/VEGF modulation) within an integrated regulatory network [17,18].

These miRNAs may not only act individually but also exert synergistic effects when delivered in combination via EVs, thereby amplifying pro-angiogenic signaling. Given their stability within EVs and biological activity in recipient cells, EV-associated miRNAs are promising candidates for next-generation nucleic acid-based regenerative therapies.

In this study, we focused on angiomiRs enriched in BM-MSC-derived EVs and established an in vitro model to analyze their individual and combinatorial effects on endothelial tube formation. Our aim was to clarify molecular mechanisms that may underlie the variable efficacy observed in clinical MSC therapy and to provide a stronger scientific rationale for ongoing and future applications of MSC-derived EVs in CLI.

Finally, EV engineering and miRNA-loading strategies are increasingly explored to enhance reproducibility and potency across donors and disease contexts, supporting translational development of EV-based nucleic acid therapeutics [19].

## 2. Materials and Methods

### 2.1. Cells

The human-derived cells used in this study were obtained from ethically approved sources (Lonza: Vacaville, CA, USA, PromoCell: Heidelberg, Germany), and their use complied with the supplier’s regulations and Declaration of Helsinki.

### 2.2. Cells and Cell Culture

Normal human BM-MSCs and human umbilical vein endothelial cells (HUVECs; C2517AS, Lonza,) were utilized in this study. BM-MSCs were cultured in MesenPRO RS Medium (12746-012; Gibco, Carlsbad, CA, USA) at 37.0 °C in a humidified atmosphere with 5.0% CO_2_. HUVECs were cultured in Endothelial Cell Growth Medium-2 BulletKit (CC-3162; Lonza) under the same conditions. Confluent cells were washed with phosphate-buffered saline (PBS) (−), detached using Accutase (AT104; Innovative Cell Technologies, San Diego, CA, USA), and centrifuged at 1000 rpm for 3 min at room temperature. After centrifugation, the supernatant was removed, and cell pellets were resuspended in MesenPRO RS Medium for subsequent experiments.

The BM-MSCs employed in this study were obtained from the following donors: 25-year-old male (PT-2501, Lot: 19TL281098; Lonza), 21-year-old male (PT-2501, Lot: 4F0218; Lonza), 23-year-old female (C-12974, Lot: 21TL046615; PromoCell), 72-year-old female (C-12974, Lot: 471Z022; PromoCell), 68-year-old male (C-12974, Lot: 467Z023.5; PromoCell), and 63-year-old male (C-12974, Lot: 465Z016; PromoCell).

### 2.3. Cell Culture for EV Collection

The resuspended cell pellets were seeded into sterile 150-mm culture dishes at a density of 6.0 × 10^5^ cells/dish, with viability confirmed via trypan blue staining. After 24 h, cells were washed with PBS (−), and the medium was replaced with 20 mL/dish of CTS StemPro MSC SFM Medium (A1033201; Gibco). The cells were incubated for 48 h to facilitate EV secretion.

### 2.4. EV Collection via Ultracentrifugation

Culture supernatants were centrifuged at 2000× *g* for 10 min at 4 °C. The supernatants were filtered through a 0.22-μm membrane filter (SCGPS02RE; Millipore, Darmstadt, Germany) and transferred to an ultracentrifuge tube. Ultracentrifugation was performed at 35,000 rpm for 70 min at 4 °C using Optima XE-90 Ultracentrifuge (Beckman Coulter, Brea, CA, USA). The resulting precipitate was resuspended in PBS (−) and weighed.

### 2.5. Nanoparticle Tracking Analysis (NTA)

The size and concentration of EVs were determined using a NanoSight LM10 system (Malvern Instruments Ltd., Amesbury, UK) equipped with a 405-nm blue laser and NanoSight NTA software v3.44. The EVs were illuminated by a laser, and the Brownian motion of particles was recorded in 90-s videos. Particle size was calculated using the Stokes–Einstein equation via the NTA 2.0 software. EV samples were diluted with 0.22-μm filtered PBS (−) to achieve a final particle concentration of 1 × 10^8^ to 2.5 × 10^9^ particles/mL.

### 2.6. ExoScreen Assay

The ExoScreen assay was performed as previously described [5] to detect exosomes. Ten microlitres of culture supernatant or EV samples were added to 96-well half-area white plates (6002290; PerkinElmer, Brea, CA, USA), followed by 15 μL of a mixture of anti-human CD63 antibody solid-phase acceptor beads and biotinylated anti-human CD63 antibody. The reaction mixture was incubated at 37 °C for 1 h in the dark. Subsequently, 25 µL of AlphaScreen streptavidin-coated donor beads were added without washing, and the plate was incubated in the dark for another 30 min at room temperature. The plate was then read on the EnSpire Alpha 2300 Multilabel Plate Reader (PerkinElmer) at an excitation wavelength of 680 nm and emission detection at 615 nm. Background signals from 0.22-μm filtered PBS (–) were subtracted from the readings.

### 2.7. RNA Purification

Cells cultured in 100-mm dishes were washed with PBS (–) and lysed using QIAzol Lysis Reagent (Qiagen, Hilden, Germany). Total RNA, including miRNAs, was extracted with the miRNeasy Mini Kit (217004; Qiagen) according to the manufacturer’s instructions. Syn-cel-miR-39-3p miScript miRNA Mimic 219,600 (MIMAT0000010; Qiagen) was used as a reference miRNA for correction. RNA concentrations were quantified using a NanoDrop 1000 spectrophotometer (Thermo Fisher Scientific, Wilmington, DE, USA).

### 2.8. Reverse Transcription-Quantitative Polymerase Chain Reaction (RT-qPCR)

miRNA reverse transcription was performed using the TaqMan MicroRNA Reverse Transcription Kit (4366597; Applied Biosystems, Foster City, CA, USA). Amplification was conducted with the TaqMan Universal PCR Master Mix without AmpErase UNG (4324018; Applied Biosystems), using miRNA-specific primers from the TaqMan MicroRNA Assay (4427975; Applied Biosystems). Endogenous controls included miR-16 and cel-miR-39. Data were analysed using the 2^−ΔΔCT^ method on the StepOnePlus Real-Time PCR System (Applied Biosystems). The TaqMan assay IDs for the target miRNAs were: miR-9: 000583, miR-105: 002167, miR-126: 002228, miR-135b: 002261, miR-210: 000512, miR-16: 000391, and cel-miR-39: 000200.

### 2.9. Tube Formation Assay Using Co-Culture of BM-MSCs and HUVECs

BM-MSCs (PT-2501, Lot: 19TL281098; Lonza) were seeded at densities of 7.0 × 10^5^ and 28.0 × 10^5^ cells and embedded in Matrigel (354230, Matrigel basement membrane matrix Reduced; Corning, Corning, NY, USA ) in 24-well plates. After incubation at 37 °C with 5% CO_2_ for 30–60 min, HUVECs (1.5 × 10^4^ cells/well) were seeded onto the Matrigel beds. Angiogenesis was assessed morphologically after 20–24 h and further analysed using Calcein AM Fluorescent staining (354216, Calcein AM Fluorescent dye; Corning). Images were captured with a fluorescence microscope and processed with the ImageJ Angiogenesis Analyzer software (ImageJ 1.53t; National Institutes of Health, Bethesda, MD, USA) [20,21].

### 2.10. miRNA Transfection into HUVECs

HUVECs (2.5 × 10^4^ cells/well) were seeded into 24-well plates and incubated at 37 °C with 5.0% CO_2_ for 24 h. Transfection was performed using mimic RNA (Invitrogen–Thermo Fisher Scientific, Waltham, MA, USA) and INTERFERin transfection reagent (101000028; Polyplus, Illkirch, France) following the manufacturer’s protocol. Mimic RNA sequences included miR-126-3p: UCGUACCGUGAGUAAUAAUGCG (MC12841; Ambion– Thermo Fisher Scientific, Waltham, MA, USA), miR-135b-5p: UAUGGCUUUUCAUUCCUAUGUGA (MC13044; Ambion), miR-210-3p: CUGUGCGUGUGACAGCGGCUGA (MC10516; Ambion), and a non-targeting negative control (NC) miRNA mimic (mirVana™ miRNA Mimic Negative Control 1, 4464058; Ambion), which contains a scrambled sequence that does not match any known human miRNA or mRNA sequences. The exact sequence of the NC mimic is proprietary and not publicly disclosed. After 48 h of incubation at 37 °C with 5.0% CO_2_, RT-qPCR was performed to confirm miRNA expression (as previously described in Section 2.8).

### 2.11. Tube Formation Assay with Mimic RNA-Transfected HUVECs

Transfected HUVECs were seeded at 1.6 × 10^5^ cells/well into 6-well plates and incubated for 24 h at 37 °C in 5.0% CO_2_. Mimic RNAs and transfection reagents were prepared as per the manufacturer’s protocol and added to the cells. Following transfection, the cells were incubated at 37 °C in 5.0% CO_2_ for 48 h. The transfected mimic RNAs included miR-126, miR-135b, and miR-210, along with combinations of miR-126 + miR-135b, miR-126 + miR-210, miR-135b + miR-210, and miR-126 + miR-135b + miR-210. Matrigel (300 μL/well) was added to 24-well plates and incubated at 37 °C in 5.0% CO_2_ for 30 min to 1 h. Transfected HUVECs (3.0 × 10^4^ cells/well) were seeded on Matrigel beds. Angiogenesis was assessed morphologically after 20–24 h and analysed as previously described in Section 2.9.

### 2.12. Evaluation of Angiogenic Protein Expression in Tube Formation Assay by Mimic RNA Transfection into HUVECs

The supernatant of the cultured 6 mm cell culture dish was removed, washed with PBS (-), and the PBS (-) was removed again. M-PER Mammalian Protein Extraction Reagent (78501; Thermo Fisher Scientific) with 1/100 of a protease inhibitor was added, and the cells were scraped off using a cell scraper and collected in a microtube. The cells were gently stirred at room temperature for 10 min, centrifuged at 14,000× *g* for 15 min at 4 °C, and the supernatant was collected. Protein was collected according to the manufacturer’s protocol. Protein was quantified using Qubit Assays (Q33211, Thermo Fisher Scientific) according to the manufacturer’s protocol. The expression levels of 55 angiogenesis-related proteins were evaluated using a membrane-type sandwich immunoassay (Proteome Profiler Human Angiogenesis Array Kit; R&D Systems, Minneapolis, MN, USA) according to the manufacturer’s protocol. The total protein amount of each sample was 100 µg. Proteins spotted on the membrane were visualized using a bioluminescence detector (Fusion Solo S, Vilber, France) with a 30-s exposure. Analysis of membrane spots was performed using ImageJ software. Protein array analysis was performed using conditioned medium from two independent MSC cultures (biological replicates, *n* = 2). For each sample, duplicate membranes supplied in the kit were used (technical replicates). Signal intensities were quantified with ImageJ, and the average of the duplicate membranes was taken as the value for each biological replicate.

### 2.13. Statistical Analysis

Statistical significance was determined using a one-way analysis of variance in BellCurve for Excel (Social Survey Research Information Co., Ltd., Tokyo, Japan). Multiple comparisons were performed using the Tukey post hoc test. A *p*-value < 0.05 was considered statistically significant.

## 3. Results

### 3.1. Confirmation of miRNA Expression in BM-MSC and BM-MSC-Derived EVs

#### 3.1.1. Characterization of BM-MSC-Derived EVs

The physical and biochemical characteristics of BM-MSC-derived EVs were evaluated using a NanoSight LM10 system (NanoSight Ltd., Amesbury, UK) equipped with a 405-nm blue laser and NanoSight NTA software v3.44. NTA to determine particle size and distribution. NTA revealed a predominant particle size peak at approximately 100 nm (Figure 1a). The EV yield was quantified at 2.0 × 10^3^ to 3.0 × 10^3^ particles per cell across passages (Figure 1b). ExoScreen analysis confirmed the presence of BM-MSC EV surface marker, CD63, indicating successful EV recovery (Figure 1c). These characterization experiments were performed once (*n* = 1) as validation of our isolation procedure. Taken together, these findings indicate that the quality of the BM-MSC-derived EVs aligns with the MISEV2023 standards established by the International Society for Extracellular Vesicles [9].

#### 3.1.2. miRNA Expression Analysis in BM-MSC and BM-MSC-Derived EVs Across Passages

We examined the expression of five angiomiRs—miR-9, miR-105, miR-126, miR-135b, and miR-210—associated with cellular senescence [12,14,16,22,23]. BM-MSCs were cultured to passages 4 (P4) and 7 (P7). miR-126, miR-135b, and miR-210 were detected in both BM-MSCs and EVs, whereas miR-9 and miR-105 were expressed exclusively in BM-MSCs and were not detected in EVs (Figure 2). These analyses were performed in triplicate (*n* = 3) to confirm reproducibility, although the purpose of this experiment was presence/absence validation rather than statistical comparison. Based on these results, miR-126, miR-135b, and miR-210 were selected for subsequent analyses.

### 3.2. miRNA Variation in BM-MSCs

RT-qPCR analysis revealed age-dependent differences in the expression of several miRNAs (such as miR-126, miR-135b, and miR-210) in BM-MSCs. In particular, BM-MSCs derived from aged donors tended to show lower expression levels of these miRNAs compared to those from young donors. Furthermore, noticeable inter-individual variability was observed even within the same age group (Figure 3). Each donor-derived BM-MSC sample was analyzed in triplicate PCR runs to ensure reproducibility, although the purpose of this experiment was to assess age-related and donor-specific differences rather than to perform statistical comparisons. These findings indicate that miRNA expression in BM-MSCs is influenced not only by chronological aging but also by donor-specific biological factors. Although it would be ideal to assess EV-associated miRNAs from elderly donors, this was not feasible due to their low proliferative capacity, making it impractical to obtain sufficient EVs.

### 3.3. Tube Formation Assay Using Co-Culture of BM-MSCs and HUVECs

In the tube formation assay using co-culture of BM-MSCs and HUVECs, angiogenesis—as measured by the number of junctions—increased significantly with higher BM-MSC numbers. The 28 × 10^3^ cell group showed a significant increase compared to both the 7.0 × 10^3^ group (*p* < 0.05) and the control group (*p* < 0.01), with clearly enhanced vascular-like structures observed morphologically (Figure 4). These experiments were performed with BM-MSCs derived from three young donors (in their 20s), each analyzed in triplicate (*n* = 3), to ensure reproducibility. These results suggest that BM-MSCs possess the ability to promote angiogenesis in vitro.

### 3.4. Tube Formation Assay in HUVECs Following Mimic RNA Transfection

In the tube formation assay following mimic RNA transfection into HUVECs, miR-126, miR-135b, and miR-210 each significantly enhanced angiogenesis compared with NC (number of junctions; *p* < 0.05 or *p* < 0.01). Among these, the combination of miR-126 and miR-135b showed the strongest pro-angiogenic effect (*p* < 0.001), and the triple combination (miR-126 + miR-135b + miR-210) also resulted in a significant increase (*p* < 0.01). In contrast, the combinations of miR-126 + miR-210 and miR-135b + miR-210 exhibited relatively weaker effects compared to other groups, and no additional enhancement was observed (Figure 5). Data represent biological replicates from three independent donors (*n* = 3), with one transfection and tube formation assay performed per donor.

### 3.5. Protein Analysis in Tube Formation Assay by Mimic RNA Transfection into HUVECs

Protein expression of angiogenesis-related factors (VEGF, FGF, Endoglin, uPA) was analyzed by antibody arrays using conditioned medium from two independent MSC cultures (*n* = 2). Co-transfection of miR-126, miR-135b, and miR-210 (ALL) significantly increased the expression of angiogenesis-related proteins, including VEGF, FGF, and Endoglin, compared to the NC group (*p* < 0.05). Dual transfection with miR-126 and miR-135b also upregulated these factors (VEGF, uAP, and Endoglin) relative to NC; however, no further enhancement was observed compared to the triple combination (Figure 6).

## 4. Discussion

This study demonstrated that multiple combinations of miR-126, miR-135b, and miR-210 play an important role in promoting angiogenesis. It should be noted, however, that the results of this study are not based on in vivo experiments using MSC-derived EVs but rather on forced incorporation of specific microRNAs into vascular endothelial cells.

While miR-9 and miR-105 were detected in BM-MSCs, their absence in EVs suggests a selective packaging mechanism. Elucidating these molecular pathways may be crucial for tailoring the therapeutic potential of MSC-derived EVs. Further research is needed to fully understand the mechanisms that govern selective miRNA packaging and release to optimize therapeutic strategies for targeting ischemic tissues.

CLI remains refractory to conventional pharmacological, catheter-based, and surgical approaches. While a 2015 meta-analysis questioned the efficacy of angiogenic regenerative medicine [24], and angiogenesis gene therapy using HGF vectors was discontinued in Japan after disappointing post-marketing results, our findings suggest that variability in angiomiR expression may contribute to inconsistent outcomes.

Analysis of miRNA expression in BM-MSCs revealed that miR-126 levels are higher in younger donors, whereas miR-210 expression exhibited significant individual variability, and miR-135b was unaffected by donor age. This suggests that younger MSCs may possess greater regenerative capacity, likely attributed to higher angiomiR activity. The age-related decline in miR-126 expression may partially explain the reduced angiogenic efficacy in older patients, a phenomenon closely associated with aging and the progression of ischemic diseases, including CLI.

Tube formation assays demonstrated a dose-dependent enhancement of angiogenesis in BM-MSC and HUVEC co-cultures, with higher MSC numbers (28 × 10^3^ cells) significantly promoting angiogenesis. This aligns with prior studies highlighting the pro-angiogenic effects of MSCs, primarily mediated through secreted factors, including miRNAs [7,8]. These findings underscore the paracrine role of EVs in MSC-mediated regeneration. These findings further support the rationale of our ongoing clinical trial using intramuscular BM-MSC transplantation in patients with CLI.

Transfection of HUVECs with miR-126, miR-135b, and miR-210, both individually and in combination, significantly enhanced tube formation. Notably, the miR-126 + miR-135b combination and the triple combination of miR-126, miR-135b, and miR-210 exhibited the most pronounced effects, suggesting interactions among these miRNAs. Interestingly, no significant differences were observed among the All group, the miR-126 group, the miR-135b group, the miR-210 group, and the miR-126 + miR-135b group, suggesting that these treatments may elicit a similar maximal pro-angiogenic effect through convergent molecular pathways. Although in our study the inclusion of miR-210 did not provide additional enhancement beyond the effect of miR-126 + miR-135b, we cannot exclude the possibility that miR-210 exerts context-dependent effects on angiogenesis. Indeed, while many studies have reported pro-angiogenic roles of miR-210 under hypoxic conditions [16], others have shown that sustained or excessive miR-210 expression may impair endothelial function and attenuate angiogenesis [25]. These findings suggest that the effect of miR-210 may vary depending on cellular context, dosage, and environmental conditions.

Furthermore, the total number of EVs secreted by MSCs and the miRNA cargo loading per vesicle may reach a saturation threshold, limiting further enhancement of angiogenic activity. Indeed, tube formation assays with human endometrial MSC-derived exosomes showed a clear plateau in pro-angiogenic effect above 100 µg/mL, with 150–200 µg/mL failing to increase tube formation—consistent with saturation of exosome uptake or cargo capacity [26]. Moreover, the biogenesis machinery that sorts miRNAs into exosomes imposes an upper limit on cargo packaging efficiency, so overexpression beyond this point does not yield additional functional gains [6]. Finally, assuming that EV-associated miRNAs drive the pro-angiogenic effect, these findings underscore the potential of developing engineered exosomes—produced via genetic technologies such as lentiviral vectors to load specific target miRNAs—as a more precise therapeutic modality rather than relying on bulk conditioned media or simple miRNA overexpression cocktails [27,28]. In particular, engineered EVs could overcome current limitations of donor variability and age-related decline in angiomiR expression by ensuring consistent and targeted miRNA delivery. Moreover, scalable manufacturing platforms for EV engineering and miRNA loading (e.g., electroporation, lipid-based transfection, or vector-mediated approaches) are being developed, which may facilitate translation from bench to bedside. By combining our findings with these technologies, future therapeutic strategies could be tailored to achieve reproducible efficacy in diverse patient populations with CLI.

Particularly, miR-126 plays a pivotal role in vascular stability and growth by targeting EVH1 domain-containing protein 1 (Spred1), a Sprouty-related protein, and PIK3R2, a regulatory subunit of PI3K [12,29]. miR-126 promotes VEGF and other growth factor signaling because Spred1 and PIK3R2 are negative regulators of cell signaling cascades and affect the MAPK and PI3K signaling pathways, respectively. Notably, miR-126 targets multiple pathways, including MAPK, PI3K, and PIK3R2. As targeting multiple signaling pathways is thought to fine-tune angiogenic responses, miR-126 may play a role in regulating the relationship between miR-135b and miR-210, both of which are involved in the HIF-1 pathway. The inhibition of miR-126 has been shown to reduce capillary density in the gastrocnemius muscle in a mouse model of hindlimb ischemia [13]. While limited studies address miR-135b, overexpression of MSCs increases Fst expression, a regulator of activin A, a key protein in inflammation, and VEGF [15,30].

PTPN1, suppressed by miR-210, has been shown to negatively regulate the activation of the VEGF receptor VEGFR2 and stabilize cell–cell adhesions by reducing the tyrosine phosphorylation of vascular endothelial cadherin [31]. These results, derived from our in vitro tube formation assay and protein analysis (Figure 6), provide mechanistic support for the therapeutic effect observed in our clinical study, which involves intramuscular administration of BM-MSCs for CLI. Moreover, while VEGF was the most strongly induced protein, we also observed significant upregulation of FGF and Endoglin, suggesting potential involvement of distinct signaling pathways. Although cross-talk with VEGF-related pathways cannot be excluded, FGF and Endoglin may reflect alternative or complementary mechanisms contributing to angiogenesis. Given that this conclusion is based solely on Figure 6 data, it is important to clearly distinguish our experimental findings from generally known mechanisms.

These findings highlight the potency of combinatorial miRNA-based therapies for treating ischemic diseases by simultaneously targeting multiple angiogenic pathways. BM-MSC-derived EVs enriched with these miRNAs could represent a non-invasive and potentially effective therapeutic strategy for CLI, a condition characterised by limited treatment options and high morbidity and mortality. Leveraging miRNA-loaded EVs to promote angiogenesis opens a compelling avenue in regenerative medicine.

To the best of our knowledge, this is the first study to explore the role of combinatorial miRNAs secreted by MSCs in angiogenesis, providing novel insights into their potential for therapeutic application. However, significant challenges remain, particularly in achieving consistent and selective miRNA packaging into EVs. While this study highlights the age-related differences in miRNA expression, the small sample size and limited donor age range restrict the generalisability of the findings. A broader cohort would be necessary to confirm the influence of donor age on miRNA profiles and angiogenic potential. This study exclusively used BM-MSCs obtained from commercially available sources, which may not fully represent the variability seen in primary cells derived from patients with diverse clinical conditions, such as CLI or other comorbidities. Although miR-126, miR-135b, and miR-210 combinations enhanced angiogenesis, their interactions and precise molecular mechanisms were not fully elucidated. This limits the ability to design targeted therapeutic strategies leveraging these miRNA combinations. While the in vitro data are promising, rigorous validation of in vivo CLI models is essential to establish the clinical utility of BM-MSC-derived EVs. By addressing these limitations, future research can refine the therapeutic application of BM-MSC-derived miRNAs and EVs for use in angiogenic regenerative medicine.

This study has several limitations. First, all experiments were conducted in vitro using HUVECs and BM-MSC-derived extracellular vesicles, and, thus, the results may not fully represent in vivo angiogenesis or clinical outcomes. Second, the number of MSC donors was limited, and all cells were obtained from commercial sources. Therefore, donor variability, including age as well as patient-related conditions such as diabetes or kidney disease, was not fully represented, which may limit the generalizability of our findings to clinical populations. Third, although our findings highlight the effects of miR-126, miR-135b, and miR-210, the mechanisms underlying the selective loading of these miRNAs into EVs remain unclear. Therefore, further in vivo validation and mechanistic studies are required before these findings can be translated into clinical application.

## 5. Conclusions

This study underscores the potential of BM-MSC-derived EVs, particularly those enriched with miR-126, miR-135b, and miR-210, to promote angiogenesis through modulation of multiple angiogenic pathways. The observed enhancement in tube formation in our in vitro co-culture model of BM-MSCs and HUVECs reflects the translational relevance of our clinical approach to treating CLI. Notably, protein expression analysis revealed that, beyond VEGF, factors such as FGF and Endoglin were also significantly upregulated, suggesting the involvement of both VEGF-dependent and independent signaling mechanisms. These findings highlight the therapeutic promise of combinatorial miRNA strategies while also emphasizing the need for further studies to optimize EV-mediated delivery and confirm efficacy in vivo.

## Figures and Tables

**Figure 1 biomedicines-13-02353-f001:**
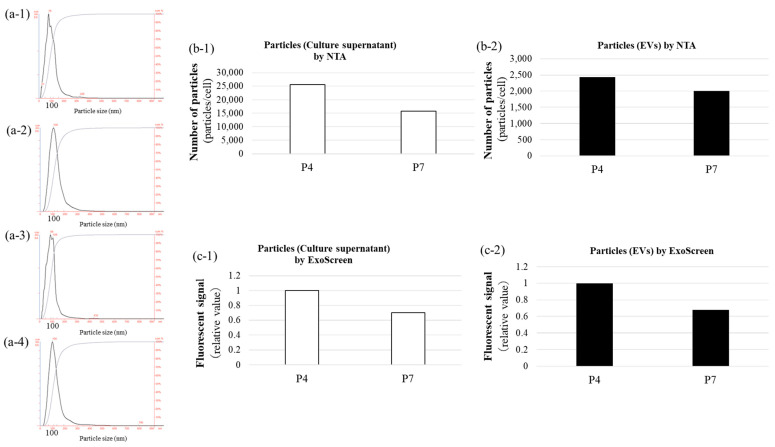
Characterisation of BM-MSC-derived EVs. BM-MSC-derived EVs were characterised by particle size, distribution, and surface marker expression. NTA confirmed a particle size peak around 100 nm. Particle numbers per cell ranged from 1.5 × 10^4^ to 2.5 × 10^4^ in culture supernatant and 2.0 × 10^3^ to 3.0 × 10^3^ for EVs. ExoScreen confirmed EV recovery based on CD63-specific luminescence (*n* = 1). Black bars represent purified EV fractions, and white bars represent BM-MSC culture supernatant. (**a-1**–**a-4**): NTA profiles showing representative particle size distributions with a peak around 100 nm, characteristic of EVs. (**b-1**): Relative particle number per cell in culture supernatant determined by NTA (white bars). (**b-2**): Relative particle number per cell in purified EV fractions determined by NTA (black bars). (**c-1**): Luminescence intensity per cell for the EV surface marker CD63 in culture supernatant measured by ExoScreen (white bars). (**c-2**): Luminescence intensity per cell for the EV surface marker CD63 in purified EV fractions measured by ExoScreen (black bars).

**Figure 2 biomedicines-13-02353-f002:**
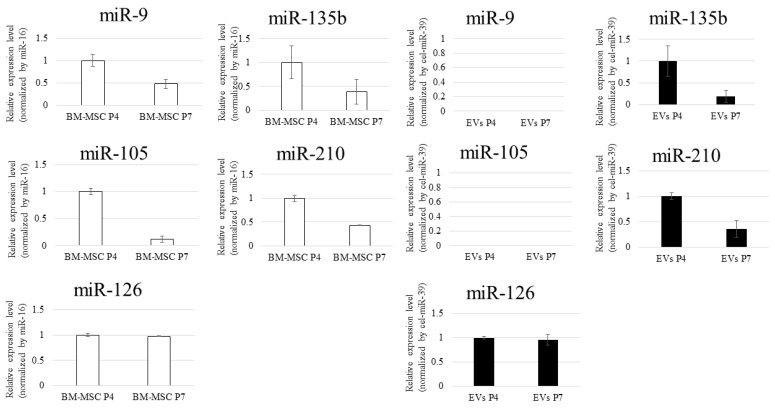
miRNA expression analysis in BM-MSCs and their EVs. RT-qPCR was used to compare the expression of miRNAs in BM-MSCs and their derived EVs. Five miRNAs (miR-9, miR-105, miR-126, miR-135b, and miR-210) were detected in BM-MSCs, while only three (miR-126, miR-135b, and miR-210) were consistently present in the EVs, whereas miR-9 and miR-105 were expressed exclusively in BM-MSCs and were not detected in EVs. Therefore, this study focused on these three miRNAs to investigate EV-mediated functional effects. miRNA expression in BM-MSCs was normalized to miR-16, and that in EVs was normalized to cel-miR-39. Data are shown as mean ± SEM. White bars represent BM-MSCs, and black bars represent EVs.

**Figure 3 biomedicines-13-02353-f003:**
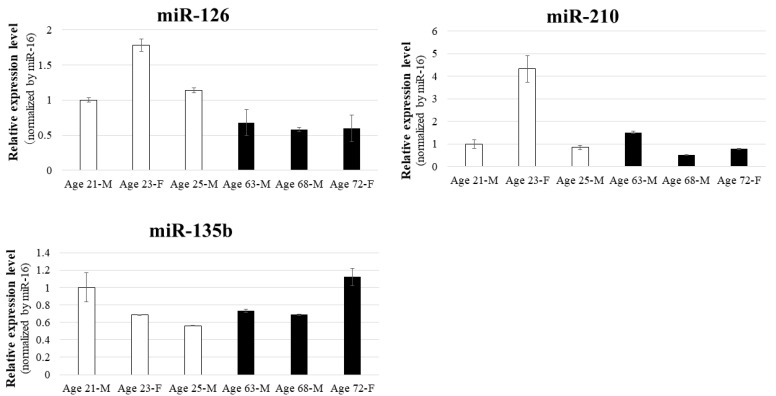
Age- and donor-dependent variability in miRNA expression in BM-MSCs. RT-qPCR was conducted to compare miRNA expression levels in BM-MSCs derived from young and aged donors. Expression was normalized to miR-16. Several miRNAs (e.g., miR-126, miR-135b, miR-210) showed differences between age groups and among individuals within the same group. White bars represent young donors (ages 21–25), and black bars represent aged donors (ages 63–72).

**Figure 4 biomedicines-13-02353-f004:**
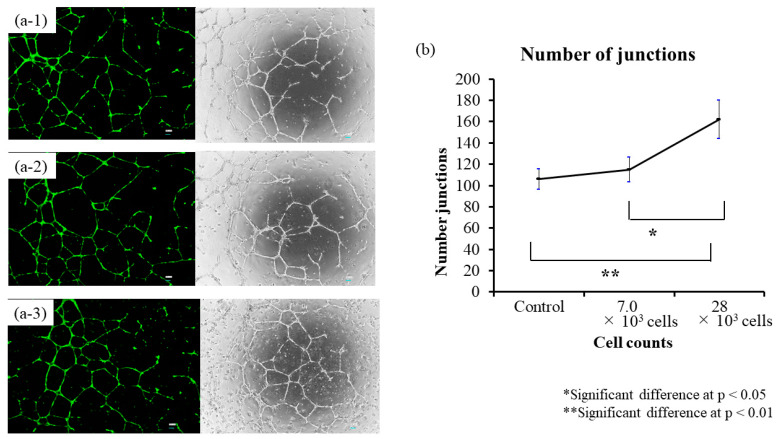
Tube formation assay using co-culture of BM-MSCs and HUVECs. BM-MSCs were co-cultured with HUVECs to evaluate dose-dependent angiogenic effects. The 28 × 10^3^ cell group showed significantly increased junction numbers compared to the control and 7.0 × 10^3^ cell groups (*p* < 0.01 and *p* < 0.05, respectively), with enhanced vascular morphology observed. (**a**) Phase-contrast images of each group: (**a-1**): Control, (**a-2**): 7.0 × 10^3^ cells, (**a-3**): 28 × 10^3^ cells. (**b**) Quantitative analysis of angiogenesis (number of junctions). Experiments were performed with BM-MSCs derived from three young donors, each analyzed in triplicate (*n* = 3).

**Figure 5 biomedicines-13-02353-f005:**
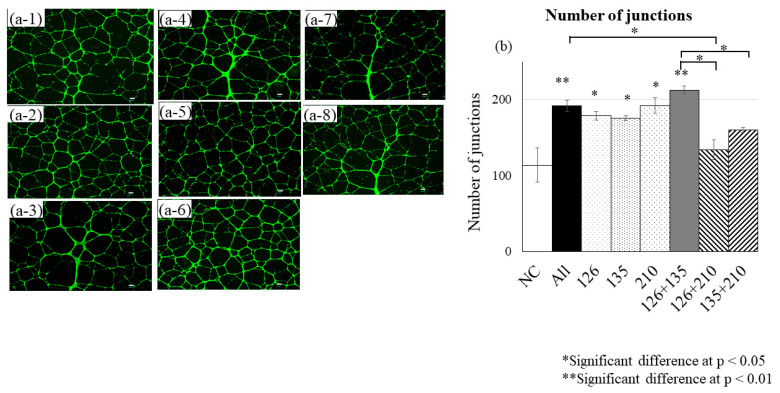
Tube formation assay in HUVECs following mimic RNA transfection. Quantitative analysis of tube structure formation 24 h after transfection. miR-126, miR-135b, and miR-210 were tested individually and in combination. (**a**) Phase-contrast/fluorescence images of tube formation: (**a-1**) NC, (**a-2**) All, (**a-3**) miR-126, (**a-4**) miR-135b, (**a-5**) miR-210, (**a-6**) miR-126 + 135b, (**a-7**) miR-126 + 210, (**a-8**) miR-135b + 210. (**b**) Quantification of junctions. Data represent biological replicates from three independent young donors (*n* = 3), with one transfection and tube formation assay performed per donor. NC: negative control mimic (4464058, mirVana™ miRNA Mimic Negative Control 1; Ambion–Thermo Fisher Scientific). Green fluorescence indicates endothelial tube-like structures. Black, white, grey, and patterned bars in (**b**) represent different experimental groups as labelled (NC, All, miR-126, miR-135b, miR-210, and their combinations).

**Figure 6 biomedicines-13-02353-f006:**
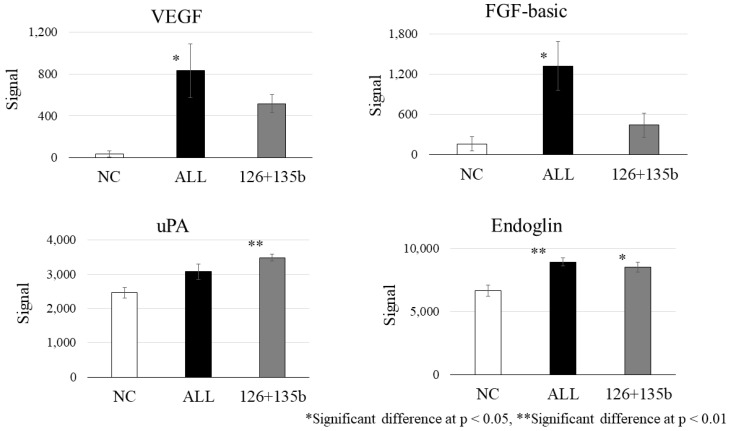
Protein expression analysis in HUVECs after miRNA mimic transfection. Expression levels of angiogenesis-related proteins (VEGF, FGF, uPA, Endoglin) were analyzed following transfection. The *x*-axis labels represent the following groups: NC (negative control), ALL (combined transfection of miR-126, miR-135b, and miR-210), and 126 + 135b (dual transfection). White bars represent NC, black bars represent ALL, and gray bars represent 126 + 135b. Conditioned medium from two independent MSC cultures was analyzed (biological replicates, *n* = 2); each sample was assayed on duplicate membranes (technical replicates), and the averaged values were used. Data are shown as mean ± SEM.

## Data Availability

All data generated or analysed during this study are included in this published article.

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
