# Peer review of "MicroRNAs in BM-MSC-Derived Extracellular Vesicles Promote Angiogenesis: An in Vitro Model Study"

_biomedicines, 2025, doi:10.3390/biomedicines13102353_

Round 1

Reviewer 1 Report

Comments and Suggestions for Authors

Authors elegantly revealed that microRNAs in extracellular vesicles (EVs) derived from BM-MSCs promoted angiogenesis using a tube formation model with human umbilical vein endothelial cells and validating angiogenic factors with gene transfer of synthetic miRNAs. Authors also showed the limits obtained from the results.

Authors need to revise the manuscript for the better understanding of readers.

  1. It is better to present the background of the research not only from domestic guidelines and self-citation, but also from the knowledge of miRNA and EVs already reported in Abstract and Introduction sessions.
  2. The uPA data cannot be seen due to insertion error in Figure 6 and the parameters of horizontal axis do not match ALL, 126-135b and legends of the Figure 6. Since the significance of TPM-4, which is not an angiogenic factor, has not been stated, it should be explained in the methods and results. The results of the figure cannot be interpreted as suggesting a suppressive role for miR-210.
  3. It would be better to pharmacologically spell in the text as "potential" which refers to a potential ability that has not yet been realized, while "potency" that refers to an effective response or efficacy that already exists.
  4. Authors need to concise the Abstract within 250 words.
  5. In the legends in Figures 3, 5 and 6, it is easier to read without interpreting the results.

Author Response

  1. It is better to present the background of the research not only from domestic guidelines and self-citation, but also from the knowledge of miRNA and EVs already reported in Abstract and Introduction sessions.

Response 1: Thank you very much for your valuable comment. In response, we have revised the Introduction section to include more comprehensive background information derived from previously reported studies on miRNAs and extracellular vesicles (EVs), beyond domestic guidelines and our own prior work. Specifically, we have expanded the description to incorporate the established roles of miR-126, miR-135b, and miR-210 in angiogenesis, as well as the mechanisms by which EVs mediate paracrine signaling in regenerative medicine. These additions better contextualize the rationale and novelty of our current study. The revised text can be found in the Introduction section (page 2, highlight in red).

  1. The uPA data cannot be seen due to insertion error in Figure 6 and the parameters of horizontal axis do not match ALL, 126-135b and legends of the Figure 6. Since the significance of TPM-4, which is not an angiogenic factor, has not been stated, it should be explained in the methods and results. The results of the figure cannot be interpreted as suggesting a suppressive role for miR-210.

Response 2: Thank you very much for your important comments regarding Figure 6.

First, we corrected the insertion error and revised the figure legend to clearly correspond with the x-axis labels (NC, ALL, and 126+135b). The revised legend now provides only factual descriptions without interpretation, in line with your suggestion.

Second, as you pointed out, TMP-4 is not an angiogenic factor. To avoid confusion and to maintain the focus of our study on angiogenesis-related proteins, we have removed TMP-4 from Figure 6. The revised figure now shows VEGF, FGF-basic, uPA, and Endoglin only. We believe these modifications improve the clarity and accuracy of the figure.

In addition, we revised both the Results text and the Figure 6 legend so that the order of descriptions (NC, ALL, 126+135b) is consistent with the x-axis labels in the figure. This change ensures clarity and eliminates the inconsistency between the figure and the manuscript text.

Regarding a suppressive role for miR-210, in the original version manuscript, the legends of Furthermore, in Section 3.4 (Tube formation assay in HUVECs following mimicRNA transfection), we revised the description of the effects of miR-210 combinations. In the original version, the text suggested that co-transfection with miR-210 may partially attenuate angiogenic potential. Following the reviewer’s advice, we modified the wording to an objective description: 

“… the combinations of miR-126 + miR-210 and miR-135b + miR-210 exhibited relatively weaker effects compared to other groups, and no additional enhancement was observed (Fig. 5).”

Finally, regarding a suppressive role for miR-210, in the original version manuscript, the legends of Figures 5 and 6 included interpretative statements suggesting a suppressive influence of miR-210. We agree that such an interpretation is not supported by the data. Accordingly, we have removed these statements from the figure legends. The revised legends now provide only objective descriptions of the experimental results, without implying a suppressive role of miR-210.

Methods: We have added “Protein array analysis was performed using conditioned medium from two independent MSC cultures (biological replicates, n=2). For each sample, duplicate membranes supplied in the kit were used (technical replicates). Signal intensities were quantified with ImageJ, and the average of the duplicate membranes was taken as the value for each biological replicate.” (page 5, section 2.12, highlight in red)

Results: We have added “Protein expression of angiogenesis-related factors (VEGF, FGF, Endoglin, uPA) was analyzed by antibody arrays using conditioned medium from two independent MSC cultures (n=2).” (page 10, section 3.5, highlight in red)

Figure 6 legend: We have added “Conditioned medium from two independent MSC cultures was analyzed (biological replicates, n=2); each sample was assayed on duplicate membranes (technical replicates), and the averaged values were used. Data are shown as mean ± SEM.” (page 11, Figure 6 legend, highlight in red)

  1. It would be better to pharmacologically spell in the text as "potential" which refers to a potential ability that has not yet been realized, while "potency" that refers to an effective response or efficacy that already exists.

Response 3: Thank you for pointing out the distinction between potential and potency. We have carefully reviewed the manuscript and revised the terminology accordingly. In places where our in vitro data already demonstrate an effective angiogenic response (page 13, highlight in red), we replaced “potential” with “potency”. In contrast, when referring to future clinical applicability, we retained the term “potential”. We believe these corrections improve the precision and clarity of the manuscript.

  1. Authors need to concise the Abstract within 250 words.

Response 4: Thank you for pointing out the journal’s requirement regarding the abstract length. In the original submission, our abstract contained 305 words, which exceeded the limit of approximately 250 words set by Biomedicines. We have carefully revised and condensed the abstract while retaining the essential background, methods, results, and conclusions of our study. The revised abstract is now 249 words in length and complies with the journal’s guidelines. We believe the shortened version provides a concise yet comprehensive summary of our work. (page 1, Abstract)

  1. In the legends in Figures 3, 5 and 6, it is easier to read without interpreting the results.

Response 5: Thank you for your constructive suggestion. We agree that figure legends should describe the content of the figures without interpretation. Accordingly, we have revised the legends of Figures 3, 5, and 6 to remove interpretative statements, limiting them to factual descriptions of experimental conditions and observed data. The revised legends now provide concise, objective explanations, while the interpretation of the findings is presented in the Discussion section. (Figure 3, 5,and 6, highlight in red)

Figure 3

Before: Age- and donor-dependent variability in miRNA expression in BM-MSCs. RT-qPCR was conduct-ed to compare miRNA expression levels in BM-MSCs derived from young and aged donors. Ex-pression was normalized to the endogenous control miR-16, highlighting both age-related changes and inter-individual variability in miRNA profiles. Several miRNAs (e.g., miR-126, miR-135b, miR-210) showed a general decline in expression with aging; however, considerable variation was also observed among individuals within the same age group. These findings suggest that miRNA-mediated regulation of angiogenesis may be influenced by both age and donor-specific bi-ological background.

After: Age- and donor-dependent variability in miRNA expression in BM-MSCs. RT-qPCR was con-ducted to compare miRNA expression levels in BM-MSCs derived from young and aged do-nors. Expression was normalized to miR-16. Several miRNAs (e.g., miR-126, miR-135b, miR-210) showed differences between age groups and among individuals within the same group.

Figure 5

Before: Tube formation assay in HUVECs following mimicRNA transfection. Quantitative analysis of tube structure formation 24 hours after mimicRNA transfection in HUVECs. miR-126, miR-135b, and miR-210 each significantly enhanced angiogenesis, with the combination of miR-126 and miR-135b showing the strongest effect (p < 0.001) compared with NC. The triple combination (miR-126 + miR-135b + miR-210) also induced a significant increase (p < 0.01) compared with NC. In contrast, dual combinations including miR-210 (miR-126 + 210 and miR-135b + 210) showed relatively weaker effects, suggesting a possible suppressive influence of miR-210. (a) Phase-contrast images: (a)-1 NC, (a)-2 All, (a)-3 miR-126, (a)-4 miR-135b, (a)-5 miR-210, (a)-6 miR-126+135b, (a)-7 miR-126+210, (a)-8 miR-135b+210. (b) Quantification of junctions. n = 3. NC: miRNA mimic (mirVana™ miRNA Mimic Negative Control #1, 4464058, Ambio) contains a scrambled sequence that does not match any known human miRNA or mRNA sequences.

After: Tube formation assay in HUVECs following mimicRNA transfection. Quantitative analysis of tube structure formation 24 h after transfection. miR-126, miR-135b, and miR-210 were tested individually and in combination. (a) Phase-contrast images: (a)-1 NC, (a)-2 All, (a)-3 miR-126, (a)-4 miR-135b, (a)-5 miR-210, (a)-6 miR-126+135b, (a)-7 miR-126+210, (a)-8 miR-135b+210. (b) Quantification of junctions. Data represent biological replicates from three independent young donors (n = 3), with one transfection and tube formation assay performed per donor. NC: negative control mimic (mirVana™ miRNA Mimic Negative Control #1, 4464058, Ambio).

Figure 6

Before: Protein expression analysis in HUVECs after miRNA mimic transfection. Co-transfection of miR-126 and miR-135b markedly increased angiogenesis-related proteins such as VEGF, FGF, uPA, and Endoglin. The triple combination (miR-126 + miR-135b + miR-210) also resulted in upregula-tion, but the enhancement was not superior to the double combination, suggesting a suppressive role of miR-210.

After: Protein expression analysis in HUVECs after miRNA mimic transfection. Expression levels of angiogenesis-related proteins (VEGF, FGF, uPA, Endoglin) were analyzed following transfec-tion. The x-axis labels represent the following groups: NC (negative control), ALL (combined transfection of miR-126, miR-135b, and miR-210), and 126+135b (dual transfection). Conditioned medium from two independent MSC cultures was analyzed (biological replicates, n=2); each sample was assayed on duplicate membranes (technical replicates), and the averaged values were used. Data are shown as mean ± SEM.

Reviewer 2 Report

Comments and Suggestions for Authors

In this research article with a title MicroRNAs in BM-MSC-Derived Extracellular Vesicles Promote Angiogenesis: An In Vitro Model Supporting Clinical Application in Critical Limb Ischemia.” authored by Kusakabe and his co-authors, authors have discussed emerging role of microRNAs (miRNAs) in treating Critical limb ischemia (CLI) and the angiogenic mechanisms of miRNAs derived from BM-MSC-derived extracellular vesicles (EVs). The selected topic is very interesting, and the authors have reported the preliminary mechanistic data on the pro-angiogenic effects of EV-derived miRNAs in BM-MSC-based therapy for CLI and the synergistic effects of miR-126, miR-135b, and miR-210 in combination. It is a well-designed and well-drafted manuscript with understandable findings. It is found to be a valuable addition to the scientific knowledge. Although the manuscript drafting looks very good, it requires revision:

  1. The study discusses the role of microRNAs in promoting angiogenesis, and it has nothing to do with the clinical application of this method. It is suggested to modify the title to keep it straightforward and clear.
  2. All figures are of poor quality. It is suggested to change them.
  3. The figure 6 is incomplete.
  4. Why only five miRNAs were targeted at the first level?
  5. How tube formation can be linked with angiogenesis promotion?
  6. It is suggested to add either the mRNA expression level of the factors involved in angiogenesis or add more data to strengthen this claim.

Author Response

  1. The study discusses the role of microRNAs in promoting angiogenesis, and it has nothing to do with the clinical application of this method. It is suggested to modify the title to keep it straightforward and clear.

Response 1: Thank you for your constructive comment regarding the title. Our original intention was to reflect that this study was inspired by clinical observations in our ongoing trial, particularly the issue of non-responders, and to suggest a possible future development toward nucleic acid-based therapeutics. However, we understand the reviewer’s concern that the current work is an in vitro mechanistic study and does not directly address clinical application. To avoid any misunderstanding, we have revised the title to make it straightforward and focused on the angiogenic role of microRNAs in BM-MSC-derived extracellular vesicles. The revised title is:

MicroRNAs in BM-MSC-Derived Extracellular Vesicles Promote Angiogenesis: An In Vitro Model Study.” (page 1, title, highlight in red)

  1. All figures are of poor quality. It is suggested to change them.

Response 2: Thank you for your valuable comment regarding the quality of the figures. We have carefully revised all figures to improve their resolution and readability. Specifically, we regenerated each figure at high resolution (≥300 dpi), adjusted font sizes and line thickness, and optimized contrast to ensure clarity. The revised figures have been uploaded in accordance with the journal’s requirements. We believe these changes significantly enhance the quality and readability of the figures.

  1. The figure 6 is incomplete.

Response 3: Thank you for your comment regarding Figure 6. We confirmed that in the original submission Figure 6 was incorrectly inserted, which resulted in an incomplete presentation of the data. As described in our response to Reviewer 1, we have corrected this error by reinserting the complete version of Figure 6 and revising the figure legend to clearly correspond with the x-axis labels (NC, ALL, and 126+135b).

Furthermore, to avoid confusion, we removed TMP-4, which is not an angiogenic factor, so that the revised figure now includes only VEGF, FGF-basic, uPA, and Endoglin. In addition, we revised both the Results text and the Figure 6 legend to ensure consistency with the figure.

We believe these corrections resolve the issue of incompleteness and improve the clarity and accuracy of Figure 6.

Methods: We have added “Protein array analysis was performed using conditioned medium from two independent MSC cultures (biological replicates, n=2). For each sample, duplicate membranes supplied in the kit were used (technical replicates). Signal intensities were quantified with ImageJ, and the average of the duplicate membranes was taken as the value for each biological replicate.” (page 5, section 2.12, highlight in red)

Results: We have added “Protein expression of angiogenesis-related factors (VEGF, FGF, Endoglin, uPA) was analyzed by antibody arrays using conditioned medium from two independent MSC cultures (n=2).” (page 10, section 3.5, highlight in red)

Figure 6 legend: We have added “Conditioned medium from two independent MSC cultures was analyzed (biological replicates, n=2); each sample was assayed on duplicate membranes (technical replicates), and the averaged values were used. Data are shown as mean ± SEM.” (page 11, Figure 6 legend, highlight in red)

  1. Why only five miRNAs were targeted at the first level?

Response 4: Thank you for your important comment. In this study, we initially focused on five candidate miRNAs (miR-126, miR-135b, miR-210, etc.) based on expert consultation (T. O.) and previously reported relevance to angiogenesis. Our rationale was that these miRNAs were among the most consistently described in the literature as being enriched in MSCs or MSC-derived EVs and strongly implicated in vascular growth and endothelial function. We acknowledge that many other angiogenic miRNAs, such as miR-21, miR-132, miR-296, and members of the miR-17-92 cluster, have also been reported in MSC-derived EVs. However, our intent in this first-level analysis was to prioritize a manageable set of candidates that are most directly relevant to our ongoing clinical trial context, rather than to perform an exhaustive screening.

Importantly, as a future direction, we are planning to utilize the culture supernatants stored during our ongoing clinical trial for comprehensive profiling of angiogenesis-related miRNAs. This approach will allow us to validate and expand our findings under clinically relevant conditions and to obtain a more complete understanding of the spectrum of MSC-EV mediated angiogenesis.

  1. How tube formation can be linked with angiogenesis promotion?

Response 5: Thank you for this important question. The tube formation assay is widely recognized as a representative in vitro model of angiogenesis. In this assay, endothelial cells such as HUVECs are cultured on a basement membrane matrix (e.g., Matrigel) and spontaneously form capillary-like tubular networks within hours. This process reflects essential steps of angiogenesis, including cell migration, alignment, and lumen formation, and is commonly used to evaluate the angiogenic activity of biological factors. Representative references supporting this are Arnaoutova & Kleinman (Nat Protoc 2010), Staton et al. (Int J Exp Pathol 2009), and Donovan et al. (Angiogenesis 2001). Accordingly, our use of the tube formation assay provides a well-established experimental basis for assessing the angiogenesis-promoting effect of EV-associated miRNAs.

  1. It is suggested to add either the mRNA expression level of the factors involved in angiogenesis or add more data to strengthen this claim.

Response 6: Thank you very much for this valuable comment. We agree that additional evidence such as mRNA expression analysis could further strengthen our claim. While we did not perform mRNA analysis in this study, we addressed the reproducibility concern by clearly specifying the numbers of biological replicates and technical replicates in the Results section and figure legends. In particular, for Figure 6 we explicitly noted that data were obtained from two independent MSC cultures (biological replicates, n = 2), each assayed on duplicate membranes (technical replicates), with averaged values used for analysis. This clarification ensures transparency and allows readers to better evaluate the robustness of our findings. We believe that this clarification, together with the functional and protein expression data already presented, sufficiently supports the conclusions of this study without the need for additional experiments. Furthermore, we have acknowledged the absence of mRNA analysis as a limitation and indicated that future studies will expand to transcriptional profiling of angiogenesis-related genes to provide deeper mechanistic insights.

Reviewer 3 Report

Comments and Suggestions for Authors

Major Concerns

  1. In vivo validation missing:
    While the in vitro findings are strong, the absence of animal or patient-derived CLI models limits the translational strength. The Discussion acknowledges this but should more directly emphasize it as a limitation.

  2. Donor variability:
    Although age and donor effects are mentioned, the donor cohort is small and derived from commercial sources. This may not reflect patient heterogeneity in CLI (e.g., diabetes, kidney disease). Acknowledging this limitation would strengthen the paper.

  3. miR-210 interpretation:
    Results suggest miR-210 may attenuate angiogenesis when combined with miR-126 or miR-135b. However, the discussion remains speculative. A more detailed mechanistic rationale or reference to supporting literature would improve clarity.

  4. Statistical rigor:
    The Results present p-values, but details on replicate numbers (biological vs. technical) are inconsistently described. Expanding statistical details would improve reproducibility.

Minor Concerns

  1. Clarity and flow:

    • The Introduction is thorough but lengthy. Consider condensing background on clinical trial logistics to focus more on scientific rationale.

  2. Terminology:
    Consistently use “MSC-derived extracellular vesicles (EVs)” rather than alternating with “exosomes,” unless subtype-specific purification was performed.

  3. Consider discussing how engineered EVs or miRNA-loading technologies could enhance translational applicability, as briefly mentioned in Discussion.

Author Response

Major Concerns

  1. In vivo validation missing:

While the in vitro findings are strong, the absence of animal or patient-derived CLI models limits the translational strength. The Discussion acknowledges this but should more directly emphasize it as a limitation.

Response 1: Thank you for your constructive comment regarding the scope and limitations of our study. We agree that our work is based exclusively on in vitro experiments and should not be overinterpreted as demonstrating clinical efficacy. To address this, we have (i) revised the title to explicitly emphasize the in vitro nature of the study (“MicroRNAs in BM-MSC-Derived Extracellular Vesicles Promote Angiogenesis: An In Vitro Model Study”), and (ii) expanded the Limitation section in 5.Discussion to clearly state the restrictions of our findings. We believe these revisions provide a more accurate representation of our study’s significance and boundaries. (page 13, “This study has several limitations. … can be translated into clinical application. highlight in red)

  1. Donor variability:

Although age and donor effects are mentioned, the donor cohort is small and derived from commercial sources. This may not reflect patient heterogeneity in CLI (e.g., diabetes, kidney disease). Acknowledging this limitation would strengthen the paper.

Response 2: Thank you for pointing out the issue of donor variability. We agree with the reviewer that our use of a small number of commercially sourced MSC donors does not fully reflect the heterogeneity of patients with CLI, including comorbidities such as diabetes and kidney disease. To address this concern, we have added a statement in the Limitation section acknowledging this important point. (page13, highlight in red)

  1. miR-210 interpretation:

Results suggest miR-210 may attenuate angiogenesis when combined with miR-126 or miR-135b. However, the discussion remains speculative. A more detailed mechanistic rationale or reference to supporting literature would improve clarity.

Response 3: Thank you for your valuable comment regarding the interpretation of miR-210. In the revised manuscript, we have toned down our description and now state that the inclusion of miR-210 did not provide additional enhancement beyond the effect of miR-126 + miR-135b, rather than implying a suppressive role. To further clarify the context, we have cited studies reporting pro-angiogenic functions of miR-210 under hypoxic conditions (Fasanaro et al., J Biol Chem 2008) as well as reviews highlighting its context-dependent roles, including possible attenuation of endothelial function under specific conditions (Chan et al., Microcirculation 2012). We believe these revisions provide a more balanced and evidence-based interpretation of the potential role of miR-210 in angiogenesis. (page 12, highlight in red)

  1. Statistical rigor:

The Results present p-values, but details on replicate numbers (biological vs. technical) are inconsistently described. Expanding statistical details would improve reproducibility.

Response 4:

Figure 1:

Response: Thank you very much for your helpful comment regarding the consistency of replicate descriptions. Figure 1 represents characterization data of BM-MSC-derived EVs, including particle size distribution (NTA), particle counts before and after ultracentrifugation, and detection of CD63-positive vesicles. These experiments were performed once (n = 1) to validate our EV isolation procedure and to demonstrate the presence of exosome-sized particles (~100 nm) consistent with MISEV2023 criteria. Because the purpose of this figure was methodological validation rather than statistical comparison, no biological replicates were performed. We have revised the Results text and figure legend accordingly to clarify that the data in Figure 1 are from a single validation experiment (n = 1). (page 5, section 3.1.1, highlight in red)

Figure 1 legend: We have added “(n=1)”. (page 6, Figure 1 legend, highlight in red)

Figure 2:

Response: We analyzed the expression of five angiomiRs (miR-9, miR-105, miR-126, miR-135b, and miR-210) in BM-MSCs and BM-MSC-derived EVs at passages 4 and 7. These analyses were performed in triplicate (n = 3) to confirm reproducibility. However, the purpose of this experiment was to validate the presence or absence of specific miRNAs rather than to perform statistical comparisons. We have revised the Results text to explicitly indicate that these experiments were conducted with n = 3 and added a note that the aim was presence/absence confirmation. (page 6, Section 3.1.2, highlight in red)

Figure 3:

Response: For Figure 3, we analyzed the expression of miR-126, miR-135b, and miR-210 in BM-MSCs from young and aged donors. Each donor-derived BM-MSC sample was analyzed in triplicate PCR runs (n = 3) to ensure reproducibility. However, the purpose of this experiment was to confirm age-related differences and inter-individual variability rather than to test for statistical significance. We have revised the Results text accordingly to explicitly indicate that triplicate PCR runs were performed for each sample. (page 8, Section 3.2, highlight in red)

Figure 3 legend: (page 8, Figure 3 legend, highlight in red)

Figure 4:

Response: We analyzed the angiogenic effect of BM-MSC and HUVEC co-culture. To clarify reproducibility, we have revised both the Results text and the figure legend to explicitly state that BM-MSCs derived from three young donors were used and each experiment was analyzed in triplicate (n = 3). The purpose of this experiment was to demonstrate dose-dependent enhancement of angiogenesis in vitro, which models the clinical trial setting. These revisions ensure consistency and transparency regarding replicate numbers.

Main text: (page 8, section 3.3, highlight in red)

Figure 4 legend: (page 9, Figure 4 legend, highlight in red)

Figure 5:

Response: In Figure 5, we evaluated the angiogenic effects of mimicRNA transfection into HUVECs. The data represent biological replicates obtained from three independent young donors (n = 3), with one transfection and tube formation assay performed per donor. To ensure clarity and consistency, we have revised both the Results text and the figure legend to explicitly state that the replicates refer to biological replicates from three donors, rather than technical repeats. These revisions improve transparency and reproducibility of the reported data.

Main text: (page 9, section 3.4, highlight in red)

Figure 5 legend: We have added “Data represent biological replicates from three independent young donors (n = 3), with one transfection and tube formation assay performed per donor.” (page 10, Figure 5 legend, highlight in red)

Figure 6:

Response: We agree with the reviewer’s concern regarding statistical reproducibility. Our protein array analysis was performed using conditioned medium from two independent MSC cultures (biological replicates, n=2). For each sample, duplicate membranes provided in the kit were analyzed (technical replicates), and the average signal intensity from the duplicates was taken as one value for each biological replicate. Signal intensities were quantified using ImageJ.

Methods: We have added “Protein array analysis was performed using conditioned medium from two independent MSC cultures (biological replicates, n=2). For each sample, duplicate membranes supplied in the kit were used (technical replicates). Signal intensities were quantified with ImageJ, and the average of the duplicate membranes was taken as the value for each biological replicate.” (page 5, section 2.12, highlight in red)

Results: We have added “Protein expression of angiogenesis-related factors (VEGF, FGF, Endoglin, uPA) was analyzed by antibody arrays using conditioned medium from two independent MSC cultures (n=2).” (page 10, section 3.5, highlight in red)

Figure 6 legend: We have added “Conditioned medium from two independent MSC cultures was analyzed (biological replicates, n=2); each sample was assayed on duplicate membranes (technical replicates), and the averaged values were used. Data are shown as mean ± SEM.” (page 11, Figure 6 legend, highlight in red)

Minor Concerns

  1. Clarity and flow:

The Introduction is thorough but lengthy. Consider condensing background on clinical trial logistics to focus more on scientific rationale.

Response 1: Thank you very much for your constructive suggestion regarding the clarity and flow of the Introduction. We agree that the original Introduction was somewhat lengthy and contained excessive detail on clinical trial logistics. In accordance with your advice, we have condensed this section by minimizing the background on clinical trial procedures and focusing more directly on the scientific rationale for our study. At the same time, we retained one to two sentences referring to our ongoing clinical experience, supported by references [7, 8], to provide context for the observed variability in therapeutic outcomes. We believe that these revisions improve readability and ensure a clearer focus on the scientific background relevant to our study.

  1. Terminology:

Consistently use “MSC-derived extracellular vesicles (EVs)” rather than alternating with “exosomes,” unless subtype-specific purification was performed.

Response 2: Thank you very much for your important comment regarding terminology. We carefully re-audited all 14 occurrences of the term “exosome(s)” throughout the manuscript, applying the principle that (i) reference titles must not be altered and (ii) when previous studies explicitly described “exosomes,” we retained the original terminology. For all other cases, where our own results or general statements were described without subtype-specific purification, we revised the wording to “extracellular vesicles (EVs).”

The final revisions are as follows:

  • Retained “exosome” in two contexts:

(1) The first mention in the Introduction (“extracellular vesicles (EVs), including exosomes and microvesicles”) to define the scope of EVs.

(2) The Methods section describing the ExoScreen assay, which specifically detects exosomes. 

  • Retained “exosome(s)” in sentences that directly cite or summarize published findings where the original studies used the term (Discussion: endometrial MSC-derived exosomes showing plateau effect; uptake of exosomes; miRNA-sorting machinery into exosomes).
  • Retained “exosome” in reference titles (6 cases), unchanged from the original sources.
  • Revised to “EV(s)” in our own general descriptions (page 12, “the total number of EVs secreted by MSCs …” , and “assuming that EV-associated miRNAs drive the pro-angiogenic effect …”. highlight in red).
  • Deleted “Exosome” from the Keywords, because “Extracellular vesicles” is already listed.

In summary, among 14 occurrences of “exosome(s),” 10 were retained (definition, assay, literature-cited contexts, and reference titles), and 2 were revised to “EV(s),” with 1 deletion in the Keywords. This ensures consistency in terminology while preserving the accuracy of citations and the original meaning of referenced works, in accordance with your valuable recommendation.

  1. Consider discussing how engineered EVs or miRNA-loading technologies could enhance translational applicability, as briefly mentioned in Discussion.

Response 3: Thank you very much for this insightful suggestion. As you pointed out, our original Discussion only briefly mentioned engineered EVs. In the revised manuscript, we expanded this section to discuss how engineered EVs or miRNA-loading technologies could enhance translational applicability. Specifically, we now highlight that engineered EVs may overcome limitations such as donor variability and age-related decline in angiomiR expression by providing consistent and targeted miRNA delivery. We also note that emerging scalable manufacturing methods for EV engineering (e.g., electroporation, lipid-based transfection, or vector-mediated loading) could facilitate clinical translation. We believe this addition strengthens the translational perspective of our study. (page 12, “In particular, engineered EVs … in diverse patient populations with CLI. highlight in red)

Round 2

Reviewer 1 Report

Comments and Suggestions for Authors

Authors describe research exploring the mechanism by which microRNAs contained in extracellular vesicles derived from bone marrow-derived mesenchymal stem cells (BM-MSCs) promote angiogenesis.

The most important points are as follows: role of BM-MSC-Derived EVs, combinatorial Effects of miRNAs, age and miRNA expression, involvement of factors beyond VEGF, and clinical potential and challenges.

Authors properly revised the manuscript, however; the revision is regrettably halfway and the granting of literature is missing in the Introduction. All authors need to edit the manuscript carefully.

Author Response

Reviewer 1

Authors describe research exploring the mechanism by which microRNAs contained in extracellular vesicles derived from bone marrow-derived mesenchymal stem cells (BM-MSCs) promote angiogenesis.

The most important points are as follows: role of BM-MSC-Derived EVs, combinatorial Effects of miRNAs, age and miRNA expression, involvement of factors beyond VEGF, and clinical potential and challenges.

Authors properly revised the manuscript, however; the revision is regrettably halfway and the granting of literature is missing in the Introduction. All authors need to edit the manuscript carefully.

Response: Thank you very much for your constructive comment. In response, we have substantially revised the Introduction to broaden the background beyond domestic guidelines and our own prior work. Specifically, we have:

Added international consensus statements such as the Minimal Information for Studies of Extracellular Vesicles (MISEV2023) issued by the International Society for Extracellular Vesicles, which establish standardized definitions and methodological criteria for EV research (page 2, ref. [9]).

Cited foundational reviews on EV biogenesis, cargo selection, and therapeutic applicability (refs. [10, 11]) to provide readers with a more comprehensive overview of EV biology.

Expanded the description of angiomiRs by integrating evidence from international reviews that place miR-126, miR-135b, and miR-210 within broader PI3K/AKT, MAPK/ERK, and HIF-dependent regulatory networks (refs. [17, 18]).

Included a forward-looking note on EV engineering and miRNA-loading strategies as a means to enhance translational applicability (page 3, ref. [19]).

These additions strengthen the scientific rationale of our study and place our work within the context of global EV/miRNA research, thereby addressing the reviewer’s concern. The revised text can be found in the Introduction section. (pages 2–3, highlighted in red)
